# Multiple-Point Metamaterial-Inspired Microwave Sensors for Early-Stage Brain Tumor Diagnosis

**DOI:** 10.3390/s24185953

**Published:** 2024-09-13

**Authors:** Nantakan Wongkasem, Gabriel Cabrera

**Affiliations:** Department of Electrical and Computer Engineering, College of Engineering and Computer Science, The University of Texas Rio Grande Valley, Edinburg, TX 78539, USA; gabriel.cabrera02@utrgv.edu

**Keywords:** brain, tumor, sensors, meningioma, electromagnetic, microwave, metamaterials

## Abstract

Simple, instantaneous, contactless, multiple-point metamaterial-inspired microwave sensors, composed of multi-band, low-profile metamaterial-inspired antennas, were developed to detect and identify meningioma tumors, the most common primary brain tumors. Based on a typical meningioma tumor size of 5–20 mm, a higher operating frequency, where the wavelength is similar or smaller than the tumor target, is crucial. The sensors, designed for the microwave Ku band range (12–18 GHz), where the electromagnetic property values of tumors are available, were implemented in this study. A seven-layered head phantom, including the meningioma tumors, was defined using actual electromagnetic parametric values in the frequency range of interest to mimic the actual human head. The reflection coefficients can be recorded and analyzed instantaneously, reducing high electromagnetic radiation consumption. It has been shown that a single-band detection point is not adequate to classify the nonlinear tumor and head model parameters. On the other hand, dual-band and tri-band metamaterial-inspired antennas, with additional detecting points, create a continuous function solution for the nonlinear problem by adding extra observation points using multiple-band excitation. The point mapping values can be used to enhance the tumor detection capability. Two-point mapping showed a consistent trend between the S_11_ value order and the tumor size, while three-point mapping can also be used to demonstrate the correlation between the S_11_ value order and the tumor size. This proposed multi-detection point technique can be applied to a sensor for other nonlinear property targets. Moreover, a set of antennas with different polarizations, orientations, and arrangements in a network could help to obtain the highest sensitivity and accuracy of the whole system.

## 1. Introduction

Although the chance of developing a malignant or cancerous brain tumor in a person’s lifetime is less than 1%, brain cancer incidence has been increasing for the past three decades. An estimated 94,000 new primary brain tumors will be diagnosed in the United States in 2024. Of those, about 25,400 will be malignant, and around 18,760 people are predicted to die from malignant brain tumors in 2024. The average survival rate for all patients with malignant brain tumors is 36% [1,2].

A brain tumor may form in brain tissue or spread to the brain tissue from nearby tissues. As the tumor grows, it creates pressure on and may affect the function of surrounding brain tissue, which causes signs and symptoms such as headaches, nausea, and balance problems. Therefore, detecting brain tumors early is imperative to improve treatment efficacy and outcome. A combination of physical exams, imaging tests, and other tests, including a neurological exam to assess the nervous system and a biopsy to examine a sample of brain tumor cells, is used to discover brain tumors.

Several works have presented prototypes for microwave detection of the set of magnetic nanoparticles associated with model tumor tissue or thrombosis-related coagulation using a magnetic field sensor [3,4,5]. A review of magnetic nanotechnology approaches [3] for various types of sensors used to detect magnetic labels was presented. Sensitive micromagnetic sensors, referred to as MI sensors, which use the magnetoimpedance effect in amorphous wires and CMOS IC electronic circuits for sharp-pulse excitation, have been developed [4]. Magnetoimpedance thin film sensors for detecting stray fields of magnetic particles in blood vessel were also proposed [5]. These techniques could be incorporated in brain tumor diagnosis. It is important to stress that microwave devices typically require careful and precise calibration based on their sensitivity, as well as potential high noise dissipation from different components.

Brain imaging techniques, e.g., positron emission tomography (PET), magnetic resonance imaging (MRI), magnetic resonance spectroscopy (MRS), computed tomography (CT), and single-photon emission computed tomography (SPECT), for monitoring the presence of tumors within the brain region have played a significant role in brain tumor detection. Brain scans, including CT and MRI, are used to confirm tumors and the degree of malignancy, since obtaining biopsies is time-consuming in the case of a brain tumor. Out of these options, MRI, which uses radiofrequency signals with a strong magnetic field for human tissue imaging, is the most frequently used method, since it can provide detailed information related to the size, type of brain tumor, and shape [3]. The brain imaging machines are very expensive (e.g., an MRI machine costs in between USD 225,000–500,000), leading to a high service cost of approximately USD 1600–8400. Hence, an alternative low-cost brain tumor detection device will be helpful for an initial state diagnosis. Moreover, another drawback of these current modalities is that they require the use of high doses of radiation; thus, other alternative, less radiative methods, like microwave tomography, should be investigated. The specific absorption rate (SAR) is a measure of how much energy is absorbed by body tissue that depends on the electromagnetic properties of the tissue, such as permittivity and permeability. There exist recommended practices and regulations that limit the maximum acceptable amount of SAR in tissue. In the United States of America, these regulations are put forth by the Federal Communications Commission (FCC) [4], and the recommended limits indicate that the SAR must not exceed 1.6 W/kg for 1 g and 2 W/kg for 10 g of tissue [5].

Based on the low radiation doses, recently, there have been several attempts toward microwave brain tumor detection as an alternative, mainly focusing on the S band (2–4 GHz) [6,7,8,9,10] and the X band (8–12 GHz) [11,12]. A three-dimensional (3D) slot antenna, operating from 1.41 to 3.57 GHz [6] and consisting of an array of four circularly polarized patch antennas operating at 2.4 GHz [7], was designed and fabricated to detect the presence of brain tumors. In addition, 2.4 GHz microstrip patch antennas were proposed to detect a tumor inside six-layer head phantoms [8,9]. A more comprehensive study was performed on a 1.37 to 3.16 GHz metamaterial stacked antenna design and tested with a detailed 3D head model [10]. An electromagnetic band gap (EBG)-based microstrip patch antenna [11] and a Graphene-based conductor (GBC) ultra-wide-band patch antenna [12] were designed to detect human brain tumors in the X band range.

However, all these existing microwave sensors use a fixed ‘one-point’ complex permittivity value of the brain tumor, e.g., $\varepsilon_{r}=55+i40$; however, in fact, the value is at a much lower frequency of 0.5 GHz. Similarly, fixed ‘one-point’ electromagnetic property values of other head layers have been used within the frequency range of interest. This rough approximation is beneficial for a trailblazer. To improve the accuracy, in this research, we define a minimum of seven data sets of property values at 1 GHz intervals for all the head layers, including the meningioma tumor, within the frequency range of 12–18 GHz. It is important to note that based on a tumor size of 5–20 mm, a higher operating frequency is essential to ensure that the wavelength is similar or smaller than the tumor target.

In this research, we focus on meningioma tumors, the most common primary brain tumor, arising from the cranial meninges. Meningiomas are not formed within the center brain mass but can grow to displace healthy brain tissue. The meningioma tissue complex permittivity values published in July 2023 by a research group from the Department of Neurosurgery, University Hospital Centre Zagreb, Croatia and measured from 38 tissue samples in the frequency range of 0.5–18 GHz [13] are implemented in this study. A head phantom composed of seven layers of (1) epidermis and dermis, (2) hypodermis/subcutaneous tissue, (3) skull, (4) dura mater, (5) subarachnoid space/cerebrospinal fluid, (6) gray matter, and (7) white matter is created. The meningioma tumor sample sizes, varying between 5–20 mm, are placed in layers (4), (6), and (7) in our case studies.

Compact, mobile, and contactless metamaterial-inspired microwave sensors, composed of multi-band metamaterial-inspired antennas, were developed to detect and identify meningioma tumors. The sensors operate in the Ku band range (12–18 GHz), where the actual tumor measurement electromagnetic property values are currently available. First, two conventional single-band antennas, i.e., rectangular and disc-shaped, were designed and tested with the head phantom. The reflection coefficients recorded and analyzed from a patch antenna can be obtained instantaneously, reducing high electromagnetic radiation consumption. The proposed antennas were tested on the aforementioned head phantom, without and with tumors of various sizes and locations. Our study has shown that the electromagnetic responses of the head model with/without the meningioma tumor in the microwave frequency range are not linear and are not quite predictable, especially if there is only one detecting point. Next, a dual-band and a tri-band Ku metamaterial-inspired antenna, each with additional detecting points, were specifically designed to improve the tumor detection accuracy. Multiple detecting locations help to distinguish the nonlinear targets, specifying the sensors’ ability to trace meningioma brain tumors.

The content of this article begins with a discussion of the meningioma brain tumor and human head model (Section 2), followed by the design of four types of Ku band antenna (Section 3). Then, the model designed in Section 2 is tested with the four antennas, starting from single-band rectangular and disc patches and continuing with dual-band and tri-band antennas in Section 4, which includes simulation results and a discussion.

## 2. Meningioma Brain Tumor and Human Head Model

There are several parameters contributing to the sensors’ accuracy and reliability. The properties and dimensions are the main keys. If available, actual and trustworthy values are advantageous so that the interaction between electromagnetic waves and head tissues can be correctly analyzed. Next, the meningioma brain tumor and human head model used in this research will be discussed in detail.

### 2.1. Human Head Model

The head is one of the most complicated structures of the human body. Mimicking the human head is a challenging task, as the head includes many different tissues with a complex distribution [14]. Table 1 lists the head layers with a detailed layer count.

In this study, a multilayer head phantom composed of seven layers, including, starting from the outer most layer, (1) skin or epidermis and dermis, (2) fat or hypodermis/subcutaneous tissue, (3) bone or skull, (4) dura mater tissue, (5) subarachnoid space/cerebrospinal fluid (CSF), (6) gray matter, and (7) white matter, as listed in Table 2 [11], was implemented to represent the closest head model approximation based on the availability of the frequency-dependent dielectric properties and dimensions. The head layer thickness depends on multiple factors, e.g., age, gender, race, etc. The possible layer thickness range is listed in the second column of Table 2. Since the variance in human dimensions is sufficiently large to not significantly impact the choice of layer dimensions, the thicknesses from Ref. [11] were selected and used to create concentric spheres with CST Microwave Studio Suite [15]. These values are presented in Figure 1.

The frequency-dependent dielectric properties for this seven-layer head phantom model are obtained from [21] in the Ku band range (12–18 GHz; wavelength = 16.7–25 mm) and are modeled using an n^th^ order model in CST Studio Suite 2023 with points defined at 12, 13, 14, 15, 16, 17, and 18 GHz for all the layers. This frequency range is carefully selected based on the available measured brain tumor dielectric properties in the frequency range of 0.5–18 GHz [13] and typical tumor sizes, which vary between 5 and 20 mm. It is important to state that the operating wavelength of the sensors’ microwave excitation must be comparable to or smaller than the size of the detected object in order to effectively detect the target.

*Epidermis/Dermis Layer:* The outermost layer of the head phantom models the epidermal and dermal layer of the human head. This layer has an outer radius of 90 mm and has a thickness of 1 mm. The dielectric properties at the center frequency of 15 GHz (corresponding to a 20 mm wavelength) are as follows for the scalp: the real part of the relative permittivity is 26.4, and the conductivity is 13.8 S/m.

*Hypodermis/Subcutaneous Tissue Layer:* The second outermost layer models the hypodermis or subcutaneous tissue, using the SAT (subcutaneous fat) dielectric properties. At 15 GHz, the real part of the relative permittivity is 7.79, and the conductivity is 2.76 S/m.

*Skull Layer:* The human skull is composed of the braincase (neurocranium) and the facial skeleton (viscerocranium). The braincase is the majority of the bone material immediately surrounding the brain tissue, and the majority of this area is composed of the frontal bone, parietal bones, occipital bone, and temporal bones.

Here, the third outermost layer models the dielectric properties of the cranium and is modeled as a hollow sphere with a radius of 87.6 mm and a thickness of 4.1 mm. The dielectric properties are found under the skull. At 15 GHz, the real part of the relative permittivity is 6.87, and the conductivity is 3.14 S/m.

*Dura Mater Layer*: The fourth outermost layer models the dura mater of the cranial meninges. Its thickness is 0.5 mm, and the layer is modeled with an outer radius of 83.5 mm. The dielectric properties are obtained from the dura mater entry. At 15 GHz, the permittivity is 28.0, and the conductivity is 13.8 S/m.

*Subarachnoid Space/Cerebrospinal Fluid Layer*: The fifth outermost layer models the subarachnoid space and cerebrospinal fluid of the cranial meninges in a single layer using the cerebrospinal fluid entry. The hollow sphere modeling this layer is 2 mm thick and has an outer radius of 83 mm. At 15 GHz, the permittivity is 43.7, and the conductivity is 24.7 S/m.

*Gray Matter Layer:* The second innermost layer models the gray matter brain tissue as a hollow sphere with a thickness of 2.5 mm and outer radius of 81 mm. At 15 GHz, the permittivity is 31.9, and the conductivity is 16.9 S/m.

*White Matter Layer:* The innermost layer models the white matter brain tissue as a sphere with a radius of 78.5 mm. At 15 GHz, the permittivity is 23.9, and the conductivity is 12.0 S/m.

In this study, the brain layer temperatures were kept constant and uniform. However, it is noteworthy to observe the effects of the non-uniform temperature distribution of the brain parts. Investigating various temperature values on a case-by-case basis could be a focus of future detailed parametric studies.

Table 3 presents the seven-point values and the electric dispersive property graphs of the seven layers.

### 2.2. Meningioma Brain Tumor

There are more than 120 different types of brain tumors, lesions, and cysts that are differentiated by where they occur and what kinds of cells they are made of. Certain types of tumors are typically benign (noncancerous), while others are typically malignant (cancerous). Others may have a 50/50 chance of being cancerous [22].

As previously discussed, one of the main challenges to designing an effective microwave brain tumor sensor is having trustworthy data on the tumor properties and dimensions. Opportunely, in July 2023, a research group from the Department of Neurosurgery, University Hospital Centre Zagreb, Croatia, published the measured values of the meningioma tissue complex permittivity of 38 samples in the frequency range of 0.5–18 GHz [13], and these data will be our main focus. It is also important to note that if other new brain tumor data are made available, one can straightforwardly follow our research setup to design microwave brain tumor sensors based on the specific brain tumor types.

Meningioma is the most common primary brain tumor, accounting for more than 30% of all brain tumors. Meningiomas originate in the meninges, the outer three layers of tissue that cover and protect the brain just under the skull. About 85% of meningiomas are noncancerous, slow-growing tumors. Almost all meningiomas are considered benign, but some meningiomas can be persistent and come back after treatment [22].

Figure 2a presents the common locations of meningiomas, indicating the common sites of tumor growth in relationship to adjacent skull, brain, and dural reflections [23]. In our study, the meningioma is modeled as a sphere with a parametric diameter, which was initially set to 5 mm. It is modeled by displacing the gray matter layer. Two possible locations of the meningioma have been modeled. In the first location, a case is modeled where the shortest path from the meningioma to the antenna traverses only the dura mater layer and outward; this case is termed the ‘exposed case’, as shown in Figure 2b. In the second location, a case is modeled where this path traverses the gray matter layer, the subarachnoid space/CSF layer, and other outward layers; this case is termed the ‘enveloped case’, shown in Figure 2c.

In the exposed configuration, the sphere representing the meningioma is placed with its surface in contact with the inner surface of the dura mater layer. The gray matter layer is modeled as partially enveloping the meningioma. The meningioma is inserted into the subarachnoid space/cerebrospinal fluid layer to model a common location of meningiomas [23]. In the enveloped configuration, the sphere modeling the meningioma is placed in contact with the inner surface of the hollow sphere modeling the gray matter layer. A hollow sphere modeling gray matter is placed such that it completely envelopes the meningioma.

Figure 3 presents the plot of the complex permittivity of meningiomas from averaged raw data, data after smoothing, and data from the Cole–Cole model. The data from all three resources are comparable. Within the frequency range of interest, namely, 12–18 GHz, the real part of the permittivity slightly decreases linearly within the range of 39.70–32.00, while the imaginary part increases nearly linearly in the range of 22–24.35.

## 3. Ku-Band Antenna Design

An antenna with a designated operating frequency range can effectively track a target of similar or smaller dimensions to its wavelength. Based on the typical meningioma size, which varies between 5 and 20 mm, antennas operating in the Ku band (12 to 18 GHz; 16.7 mm to 25 mm), K band (18 to 26.5 GHz; 11.3 mm to 16.7 mm), and Ka band (26.5 to 40 GHz; 5.0 mm to 11.3 mm) should be ideal to trace the tumors. Since the meningioma electromagnetic properties are currently available from 0.5–18 GHz [13], Ku antennas were designed as an electromagnetic stimulus tool.

Here, we began the tumor detecting investigation using two simple single-band antennas, i.e., conventional rectangular and disc patch antennas. Then, we increased the detection capability using dual-band and tri-band patch antennas. Since the tumor and the head model electromagnetic parameters are nonlinear, the two-point and three-point mapping observations improve the tumor tracing ability [24,25,26].

### 3.1. Single-Band Ku-Band Patch Antenna

#### 3.1.1. Conventional Rectangular Patch

A conventional patch antenna was designed to generate a single passband centered at 15 GHz. The antenna was at first built using a 1 oz double-sided lossy FR-4 ($\varepsilon_{r}=4.3,\tan\delta =0.025$), with a thickness of 1.5 mm. The ground plane is square with the dimensions of 12.89 mm × 12.89 mm. The radiating patch, 7.0 mm × 10.13 mm, is placed in the middle of the FR-4 substrate and is fed at the end using a 50 Ω discrete port (0.25 mm × 3.95 mm). The inset distance, *R*, is 1 mm. Figure 4a illustrates the antenna structure, and the dimensions are listed in Table 4.

This analysis mainly uses the reflection coefficient (S_11_)_,_ which is the ratio of a reflected wave to an incident wave, to measure how much a wave is reflected from the studied models. Figure 4 illustrates the antenna along with its dimensions and the reflection coefficient (S_11_) plot, simulated using a frequency-domain solver. Four commercial substrate types, i.e., FR-4 lossy ($\varepsilon_{r}=4.3,\tan\delta =0.025$), Rogers AD 350A loss-free ($\varepsilon_{r}=3.5,\tan\delta =0$), Rogers AD 410 loss-free ($\varepsilon_{r}=4.1,\tan\delta =0$), and Rogers AD 430 loss-free ($\varepsilon_{r}=4.3,\tan\delta =0$), were investigated. With the same permittivity of 4.3, FR-4 (solid red line) and Roger 430 (dashed orange line) generate a center band at the same location at 14.81 GHz, while the center frequency ($f_{c}$) values of Rogers 350 and 450 are located at 16.35 GHz and 14.99 GHz, respectively, as expected. The relationship between the center frequency and the substrate permittivity is expressed as follows: $f_{c}=\frac{1}{2L\sqrt{\varepsilon_{0}{\varepsilon_{r}\mu }_{0}}}$. The −6 dB BW% of the FR-4 is the broadest at 5.54. Note that the antenna bandwidth is also correlated with the antenna dimensions and the substrate permittivity, which is expressed as follows: $BW\;\alpha \;\left(\frac{\varepsilon_{r}-1}{\varepsilon_{r}^{2}}\right)\;\left(\frac{W}{L}\right)\;H_{S}$. Depending on the substrate loss tangent, the antenna gain at 15 GHz is 2.039, 4.331 and 4.764 dBi for the FR-4, Rogers 410, and 430, respectively. BW% and gains are listed in Table 5.

Figure 5 presents 1D polar plots and 3D far-field plots of the three antennas. The main lobe direction of the three antennas is almost identical at 42.0° and 43.0° for the FR-4 and Rogers, respectively. The half-power or 3 dB angular beamwidth is also similar in the range of 65.8° and 67.0°. This broad beamwidth is beneficial for sensor coverage.

Even for a single antenna, marked as Ku antenna 1 and placed 5.13 cm or 2.56 λ from the head model, its half-power beamwidth of 62° can cover the whole head model, as shown in Figure 6. Note that the far-field range of this Ku antenna is 16.62 mm. This value is calculated using $d_{F}=\frac{2D^{2}}{\lambda }$, which is used for antennas physically larger than a half-wavelength, where D is the longest linear antenna dimension and λ is the operating frequency. It means that the excitation field transmitting to the head model is stable. By setting up a sensor network and adding antennas 2, 3, and 4, one can collect and monitor other scattering S-parameters, e.g., $\left(\begin{array}{ccc} S_{11} & \cdots & S_{14}\\ \vdots & \ddots & \vdots \\ S_{41} & \cdots & S_{44} \end{array}\right)$, which can be further used in imaging analysis.

#### 3.1.2. Single–Band Disc Patch

A simple patch antenna was designed to generate a single passband centered at 15 GHz. The antenna was built using a 1 oz double-sided lossy FR-4 ($\varepsilon_{r}=4.3,\tan\delta =0.025$), with a thickness of 1.6 mm. The ground plane is square with dimensions of 10 mm × 10 mm. The radiating disc patch with a diameter of 3 mm is placed in the middle of the FR-4 substrate and is fed at the end by a 50 Ω discrete port (0.20 mm × 3.60 mm). The inset width is 0.1 mm and is inserted 0.09 mm inside the disc to create a better impedance match. Figure 7 illustrates the antenna structure with dimensions and its S_11_ plot. The center frequency band is located at 14.75 GHz (−47.08 dB) with 8.68 BW%. It is also shown that a substrate with different permittivity values can be used to tune the antenna’s operating frequency. The 1D polar and 3D far-field plots of a disc antenna at 15 GHz are presented in Figure 8. The main lobe direction is 1° with the 3 dB angular width of 89.8°. The directivity is 6.21 dB. This simple, high-gain disc patch, with broad coverage and bandwidth, is perfect for achieving a normal (perpendicular) radiation pattern from the plane.

### 3.2. Dual–Band Metamaterial–Inspired Ku–Band Patch Antenna

The design strategy for obtaining two passbands in the Ku band was to create an antenna with long structures that are half or a quarter wavelength in dimension, corresponding to frequencies in the 12 to 18 GHz range. To accomplish this, antenna structures were lined up colinear to the horizontal or vertical axis. For the horizontal axis (higher frequency resonance), rectangular cuts of 0.5 mm width and 2 mm length were made on the left and right side of the patch to create multiple long horizontal elements. For the vertical axis (lower frequency resonance), the feed line and the narrow center of the patch are taken as a long vertical element. This approach resulted in the patch center vertical offset becoming a parameter for tuning one resonance, and the patch width and rectangular cut lengths were parameters used to tune the other resonance.

The proposed irrigation-like dual band patch was built using the same substrate as the single-band Ku-band patch, namely, double-sided 1 oz copper and $\varepsilon_{r}=4.3$ FR-4. The ground and the patch dimensions were 15 mm × 15 mm and 5.78 mm × 5.25 mm, respectively. The feed dimensions were 0.52 mm × 3.31 mm. There were five symmetric x-axis cuts on both left and right sides for frequency tuning purposes. The first and second bands were centered at 13.50 GHz and 16.07 GHz with the 50% transmission bandwidth at −6 dB of 3.56% and 3.36%, respectively. The directivity for the first and second bands was 6.26 dBi and 7.38 dBi, respectively, with the main lobe directions at 47.0° and 4.0°. The 3 dB angular width was 64.3° and 56.4°. The patch design with dimensions and its S11, 1D polar, and 3D far-field plots are illustrated in Figure 9.

### 3.3. Tri–Band Metamaterial–Inspired Antenna

The design strategy was to first create a multi-band enhancer using a well-known metamaterial split ring resonator (SRR) [27,28,29,30,31,32,33,34,35] and then integrate it with a broadband antenna. The combination of a set of nine SRRs and a low-profile dipole broadband antenna generated multiple passbands.

#### 3.3.1. SRR Metamaterial Band Enhancer

Nine split ring resonators (SRR) were designed to create multi passbands within the frequency range of interest. Each SRR line width and gap space between adjacent SRRs is set equally at 0.2 mm. The SRR radius values are 4.0/3.8, 3.6/3.4, 3.2/3.0, 2.8/2.6, 2.4/2.2, 2.0/1.8, 1.6/1.4, 1.2/1.0, and 0.8/0.6 mm from the outer SRR to the inner SRR, respectively. A 1.0 mm length gap was cut out from all SRRs to create broken symmetry [same SRR refs]. The SRRs were printed using 1 oz copper (0.035 mm thick). Figure 10a shows the SRR metamaterial band enhancer and (b) its S parameters. Table 6 lists the S_11_ or SZmin(1),Zmin(1) and S_21_ or SZmax(1),Zmin(1) resonances.

Note that the highlighted S_11_ resonances at 12.05, 14.55, and 17.43 GHz are the ones that will affect the additional bands when the SRRs are combined with the broadband dipole.

#### 3.3.2. Low-Profile Broadband Dipole Antenna

A low-profile broadband dipole antenna was designed to operate in the Ku band, centered at 15 GHz. The dipole was built from a copper wire with a diameter of 0.4 mm. The 50 Ω discrete port was inserted in the 1 mm gap between the 4.0 mm cylindrical wires. The S_11_ dip is centered at 14.97 GHz with a 36.67% bandwidth from 12.92 to 18.41 GHz. The main lobe direction is 0° with a 3 dB angular width of 78.6°. Since the radiation pattern is omni directional, the dipole’s gain is not high at 2.15 dBi. The dipole dimensions along with its S11, 1D polar, and 3D far-field plots are presented in Figure 11.

#### 3.3.3. Tri-Band Antenna

Nine SRR metamaterials were placed on one side of the Ku dipole antenna in the dipole radiation pattern direction. The distance between the nine SRRs and the dipole antenna (i.e., 0.5, 1.5, 3.0, 4.0, 4.5, and 5.0 mm) was investigated in order to generate three clear passbands. Based on the dominant three passbands, a distance of 3.5 mm was selected for the tumor detection investigation in the next session. Figure 12 illustrates the tri-band antenna, its S_11_ plot with varying distance parameters, and the 3D far-field plots at 14, 15, and 17.5 GHz.

## 4. Brain Tumor Microwave Sensor

The meningioma brain tumor and human head model designed in Section 2 was tested with the four antennas, starting from single-band rectangular and disc patches followed by dual-band and tri-band antennas.

### 4.1. Single-Band Antenna

Figure 13 presents the cross-sectional image of the head model placed in front of the rectangular patch antenna in the far-field range. According to their main lobe direction, the rectangular and disc patch antennas were placed 42° and 0° with respect to the normal line from the head model surface.

#### 4.1.1. Rectangular Patch

Figure 14 illustrates the S_11_ plots when the rectangular patch was used to excite a head model with various brain tumor diameters, i.e., 2, 4, 6, 8, 10, 12, 14, 16, 18, and 20 mm. There is a certain trend showing that the frequency shifts toward higher values and the S_11_ values become less negative as the tumor size increases. This trend is evident in the comparison of tumors with sizes of 2, 4, 16, and 20 mm, as shown in Figure 14a. This trend matches the results of recently published microwave sensor articles [6,7,8,9,10,11,12], which used only a fixed ‘one-point’ complex permittivity value for the head layer as well as a brain tumor in the whole frequency range. This rough approximation created a simple linear problem and solution. However, since the actual electromagnetic properties of the head layers and the tumor are in fact nonlinear, the solutions are expected to be nonlinear, where the random points are presumed. The S_11_ plot of the 10 samples are presented in Figure 14b. The S_11_ values are listed in Table 7.

#### 4.1.2. Disc Patch

Similar nonlinear results with random S_11_ dip points were observed when the head model was tested with a single-band disc antenna. The S_11_ plot of the 10 samples are presented in Figure 15. The S_11_ values are listed in Table 8.

### 4.2. Dual-Band Antenna

In this research, we proposed to add extra observation points using multiple-band excitation to create a continuous function solution for the nonlinear problem. The point mapping value can be used to channel the tumor detection capability.

Figure 16 presents S_11_ values at the two passbands. The S_11_ values within the bands at 13.24 GHz and 15.8 GHz are listed in the value order in Table 9. The S_11_ value order was repeated in all cases. The two-point mapping shows a consistent trend between the S_11_ value order and the tumor size.

### 4.3. Tri-Band Antenna

Figure 17 presents S_11_ values at the three passbands. The S_11_ values within the bands at 13.72 GHz, 15.1095 GHz, and 17.48 GHz are listed in the value order in Table 10. The S_11_ value order was random in all cases. However, three-point mapping can be used to suggest the trend between the S_11_ value order and the tumor size.

## 5. Conclusions and Discussion

Four metamaterial-inspired antennas, including two single-band antennas (a rectangular patch and a disc patch), a dual-band antenna, and a tri-band antenna, were designed in the Ku band range. We mimicked an actual human head using a seven-layered head phantom with meningioma tumors, applying actual electromagnetic parametric values. The reflection coefficients (S_11_) were recorded and analyzed instantaneously from an antenna, reducing high electromagnetic radiation consumption. Since the results showed that a single-band detecting point is not adequate to classify nonlinear tumors and head model parameters, here, we proposed to add extra observation points using multiple-band excitation to create a continuous function solution for this nonlinear problem. The point mapping value can be used to assess the tumor detection capability. A consistent trend between the S_11_ value order and the tumor size is indicated by the two-point mapping system. In addition, three-point mapping can also be employed for similar purposes. This proposed multi-detecting point technique can be implemented for other nonlinear target problems. To further improve the sensor’s sensitivity and accuracy, a set of antennas with different polarizations, orientations and arrangements in a network can be designed to collect the transmission coefficients, which can then be used to extract other electromagnetic parameters, including complex permittivity and permeability, chirality, and polarization, to name a few. This is a preliminarily simulation work on designing alternative microwave sensors using Ku-band antennas. Further tests on animals or humans will be a part of critical future work.

## Figures and Tables

**Figure 1 sensors-24-05953-f001:**
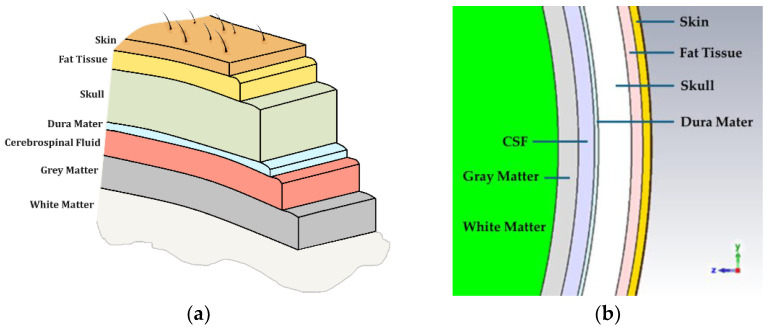
(**a**) The multilayer-head phantom and (**b**) cross-sectional model in CST.

**Figure 2 sensors-24-05953-f002:**
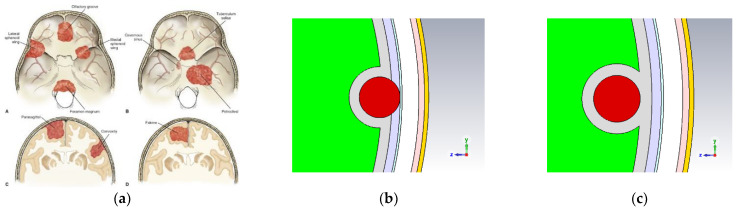
(**a**) Common locations for meningiomas. Common sites of tumor growth in relationship to adjacent skull, brain, and dural reflections; A,B: Skull base meningiomas and C,D: Falcine meningiomas attached to the dense fibrous tissue of the falx [23]. Modelling Meningioma tumor locations: (**b**) exposed and (**c**) enveloped configurations.

**Figure 3 sensors-24-05953-f003:**
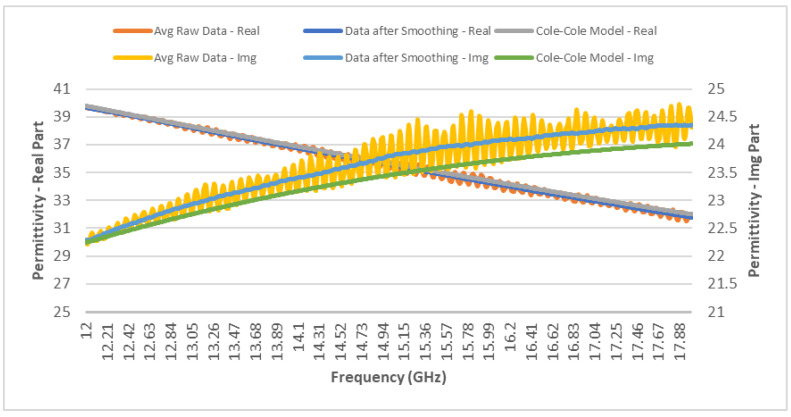
Frequency-dependent complex permittivity of meningiomas from 12–18 GHz [13].

**Figure 4 sensors-24-05953-f004:**
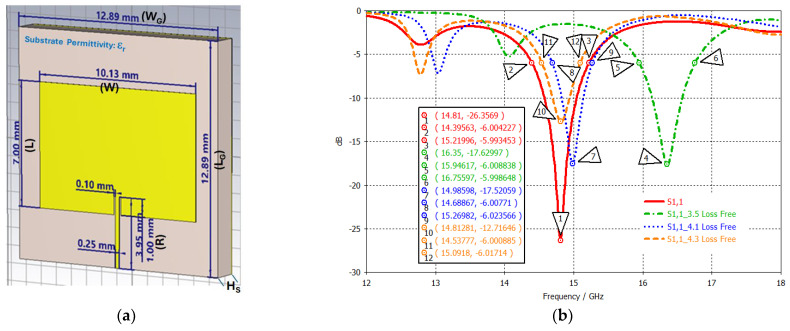
(**a**) Single–band Ku–band patch antenna and (**b**) its S_11_ plot of different substrates.

**Figure 5 sensors-24-05953-f005:**
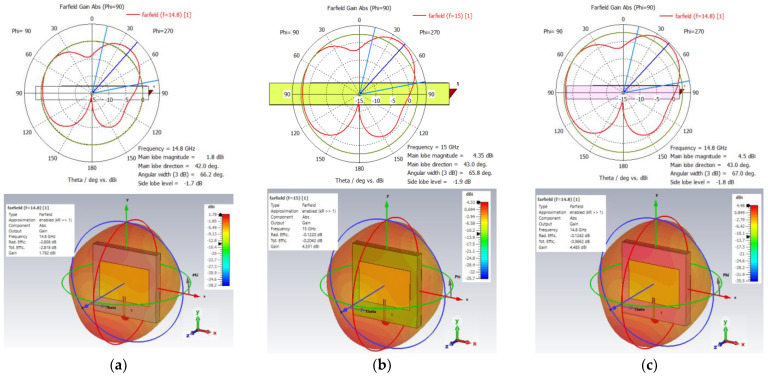
1D polar and 3D far-field plots of (**a**) FR–4, (**b**) Rogers 410, and (**c**) Rogers 430 substrate antenna.

**Figure 6 sensors-24-05953-f006:**
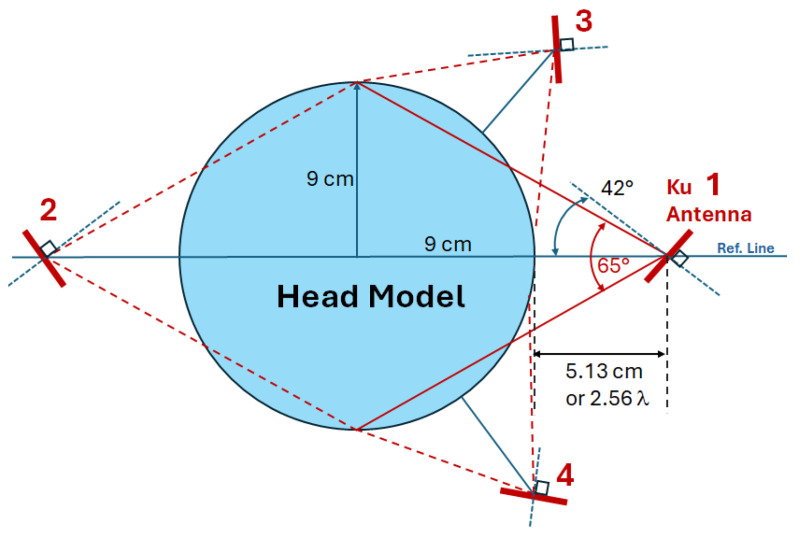
Head model and Ku-band single-band antenna setting.

**Figure 7 sensors-24-05953-f007:**
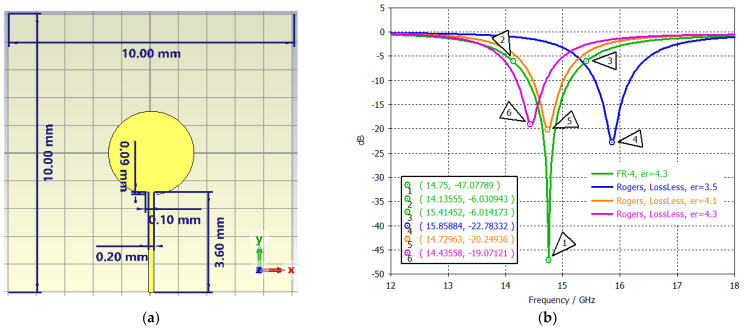
(**a**) Single–band Ku–band disc patch antenna and (**b**) its S_11_ plot.

**Figure 8 sensors-24-05953-f008:**
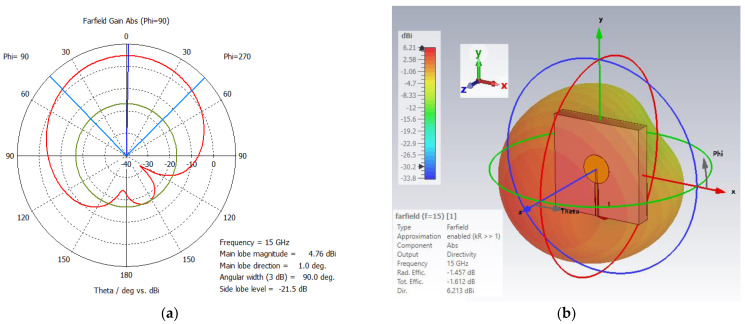
(**a**) 1D polar plot and (**b**) 3D far–field plot of a disc antenna at 15 GHz.

**Figure 9 sensors-24-05953-f009:**
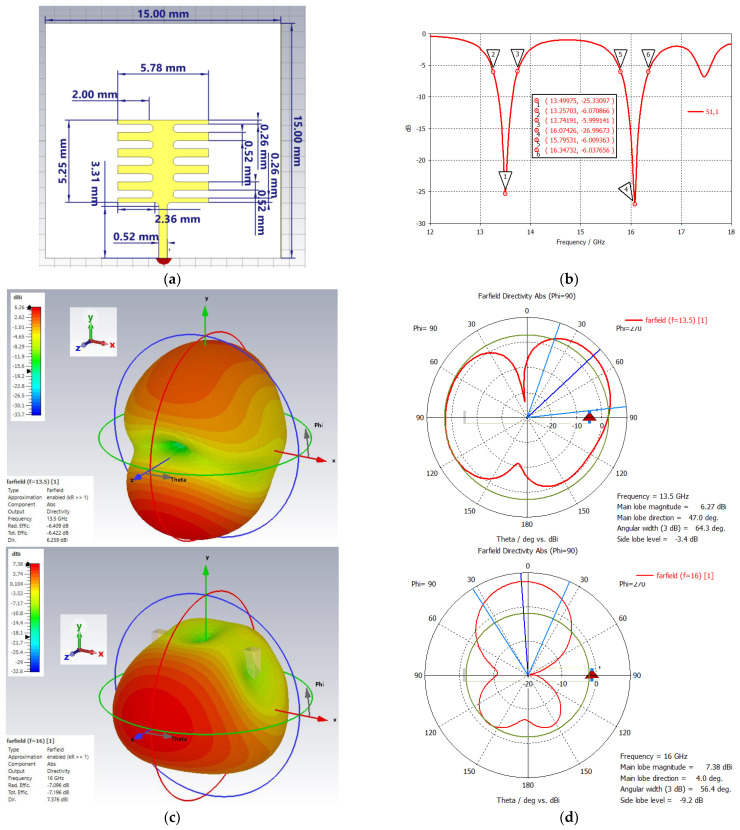
(**a**) Dual–band patch, (**b**) S_11_ plot, (**c**) 1D polar plot, and (**d**) 3D far–field plot.

**Figure 10 sensors-24-05953-f010:**
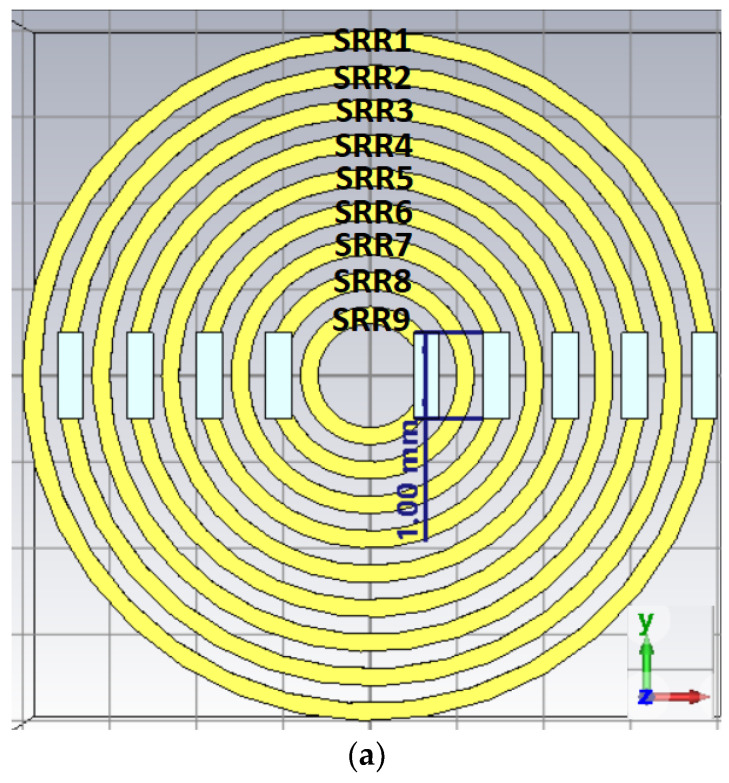
(**a**) Nine SRR metamaterial structures. (**b**) Nine SRR metamaterial S parameters.

**Figure 11 sensors-24-05953-f011:**
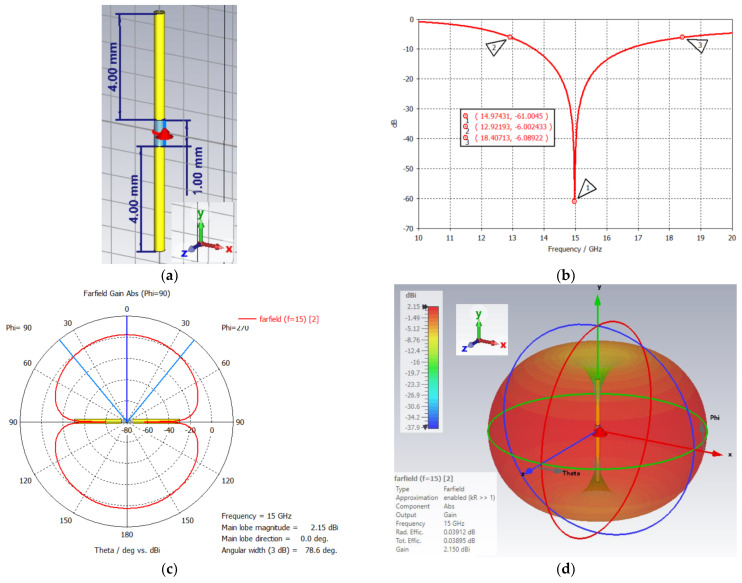
(**a**) Ku dipole and its (**b**) S_11,_ (**c**) 1D polar, and (**d**) 3D far–field plots.

**Figure 12 sensors-24-05953-f012:**
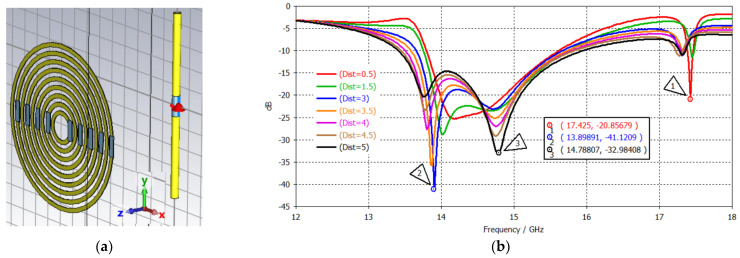
(**a**) Tri–band metamaterial-inspired antenna, (**b**) its S_11_ plot, and 3D far–field plots at (**c**) 14 GHz, (**d**) 15 GHz, and (**e**) 17.5 GHz.

**Figure 13 sensors-24-05953-f013:**
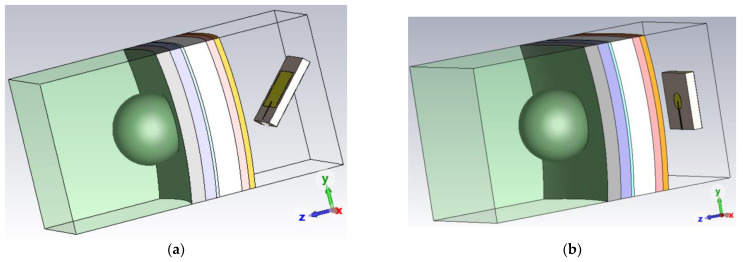
Testing the head model with a 8 mm diameter tumor using (**a**) rectangular and (**b**) disc patch antennas.

**Figure 14 sensors-24-05953-f014:**
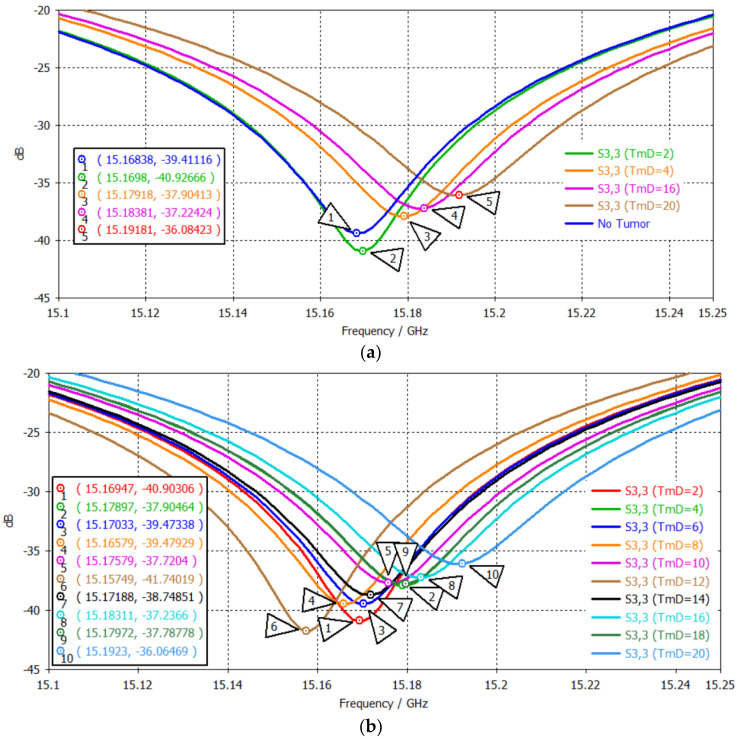
S_11_ plots when testing the rectangular patch using different brain tumor diameters: (**a**) 2, 4, 16, and 20 mm, and (**b**) 2, 4, 6, 8, 10, 12, 14, 16, 18, 20 mm.

**Figure 15 sensors-24-05953-f015:**
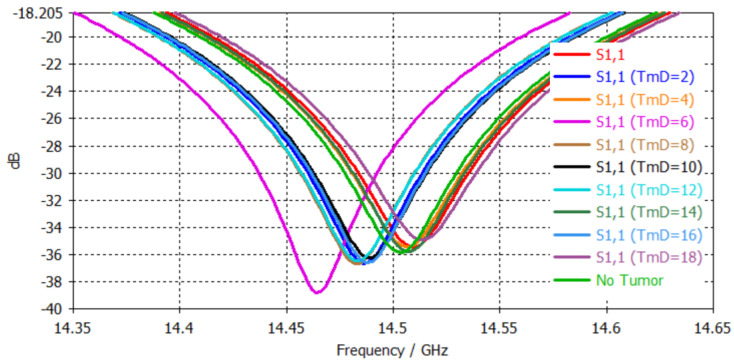
S_11_ plots when testing the disc patch using different brain tumor diameters.

**Figure 16 sensors-24-05953-f016:**
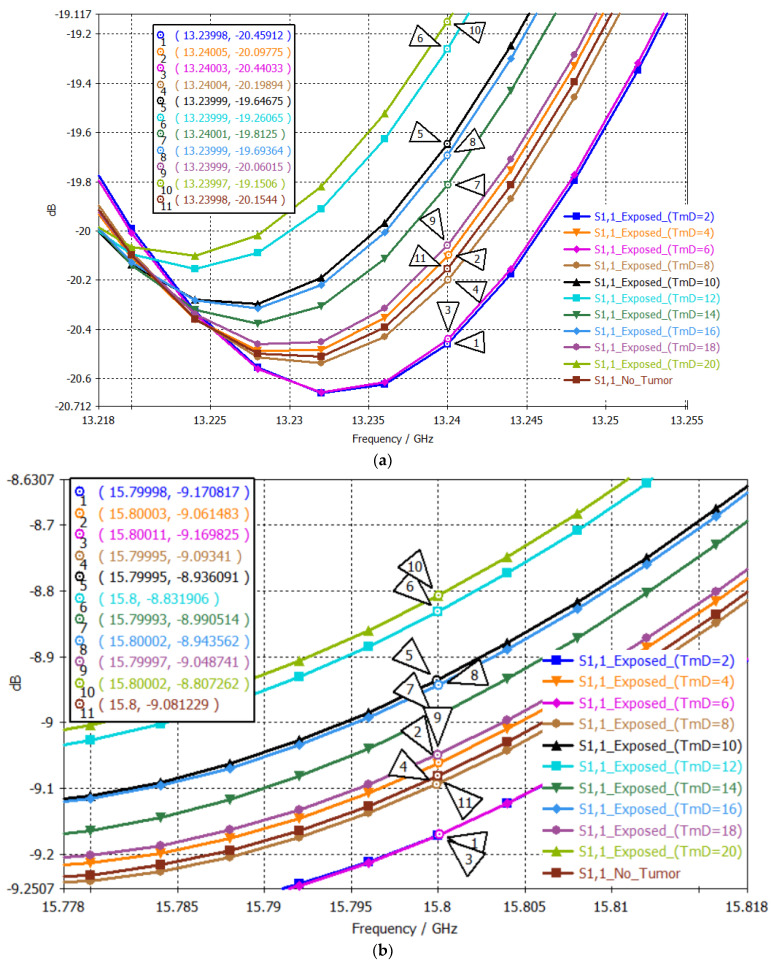
S_11_ plots when testing the dual–band patch: (**a**) 1st and (**b**) 2nd band, using brain tumors with different diameters.

**Figure 17 sensors-24-05953-f017:**
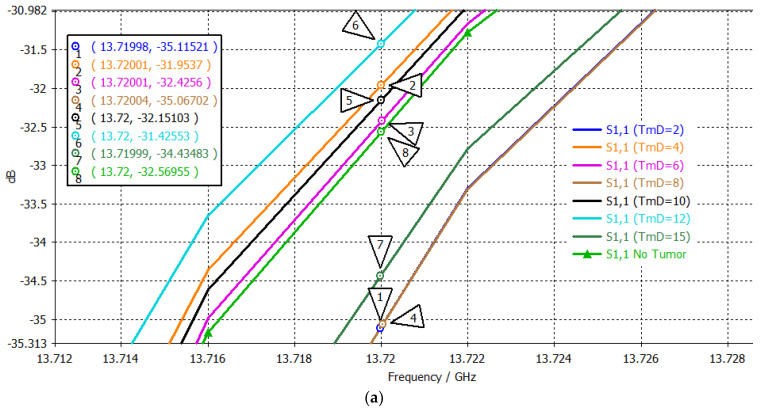
S_11_ plots when testing the tri–band patch: (**a**) 1st, (**b**) 2nd, and (**c**) 3rd band, using brain tumors with different diameters.

**Table 1 sensors-24-05953-t001:** Detailed head model.

**1. Skin**	-Scalp Skin * (1) -Subcutaneous Tissue * (2) -Aponeurosis
**2. Skull/Periosteum**	-Outer Table/Compact Bone * (3) -Diploe -Inner Table/Compact Bone
**3. Meninges**	-Dura Mater * (4) [Anatomy of the Cranial and Spinal Meninges] -Endosteal Layer -Meningeal/Fibrous Layer -Neurothelium -Arachnoid Mater -Parietal Arachnoid Layer —Subdural Layer —Central Layer -Subarachnoid Space * (5) -Leptomeningeal Trabeculae (web-like structures) -Cerebrospinal Fluid -Pia Mater -Epipial Layer -Intima Pia/Inner Layer
**4. Brain**	-Neocortex/Gray Matter * (6) -White Matter * (7)

* These components will be applied in our head model.

**Table 2 sensors-24-05953-t002:** Head layer dimensions [11,16,17,18,19,20].

	Head Layer	Thickness Range (mm) [11,16,17,18,19,20]	Thickness (mm) [11]	Radius (mm) [11]
1.	**Skin or Epidermis/Dermis**	1.10–3.25	1.0	90.0
2.	**Fat or Hypodermis/Subcutaneous Tissue**	1.40–7.00	1.4	89.0
3.	**Bone or Skull**	2.31–9.30	4.1	87.6
4.	**Dura Mater Tissue**	0.50	0.5	83.5
5.	**Subarachnoid Space/** **Cerebrospinal Fluid (CSF)**	0.44–2.40	2.0	83.0
6.	**Gray Matter**	2.50	2.5	81.0
7	**White Matter**	78.50	78.5	78.5

**Table 3 sensors-24-05953-t003:** Frequency-dependent dielectric properties of human head layers.

Layer	Electric Dispersive Properties
**1. Skin or Epidermis/Dermis** 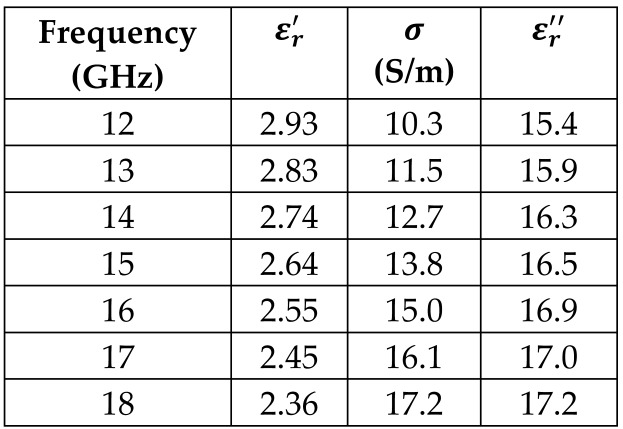	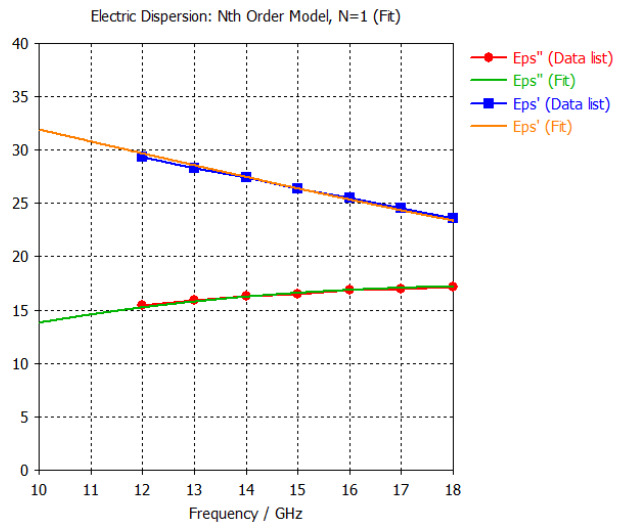
**2. Fat or Hypodermis/Subcutaneous Tissue** 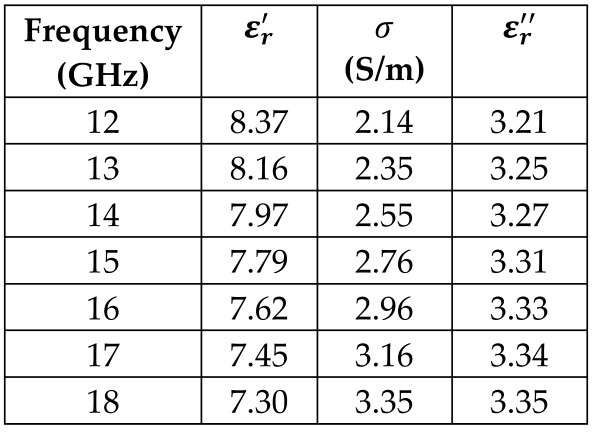	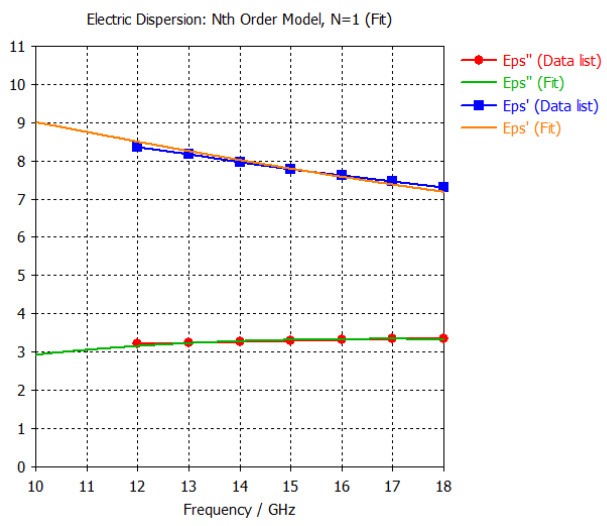
**3. Bone or Skull** 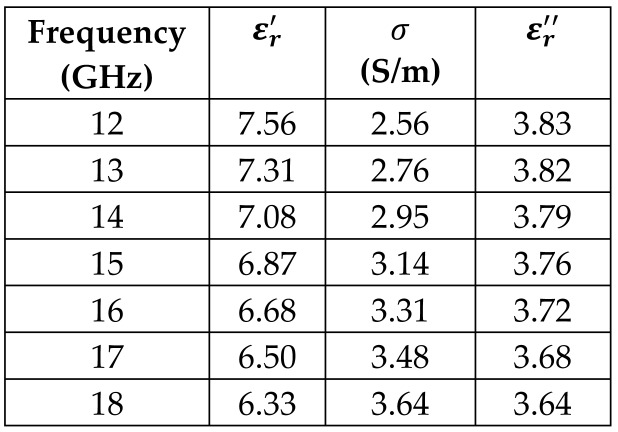	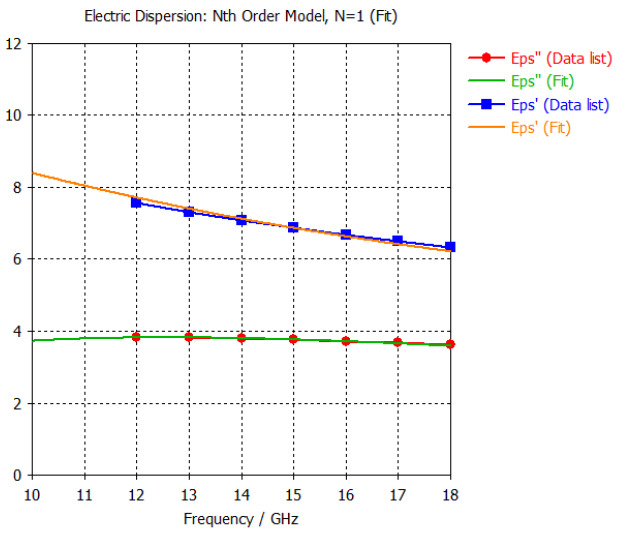
**4. Dura Mater Tissue** 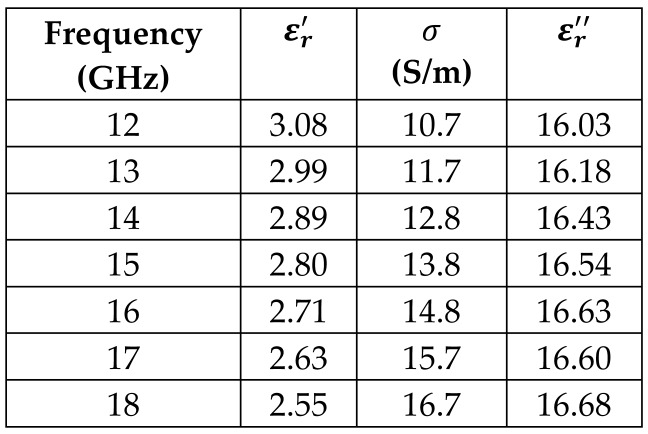	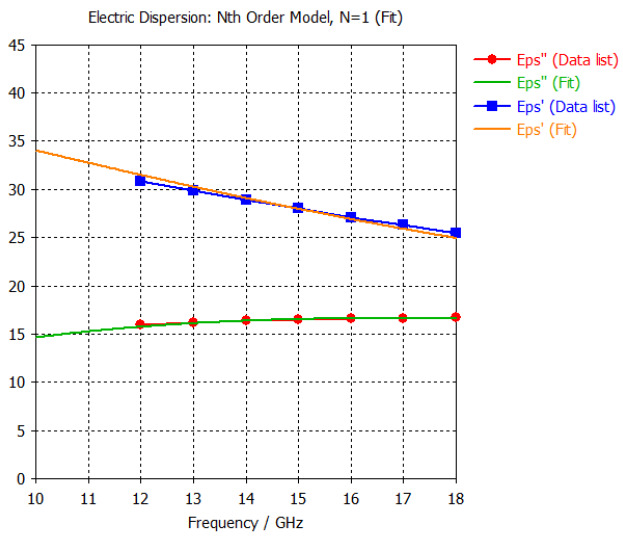
**5. Subarachnoid Space/Cerebrospinal Fluid (CSF)** 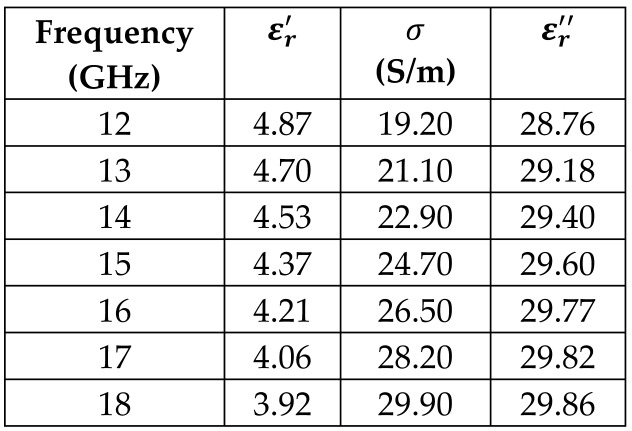	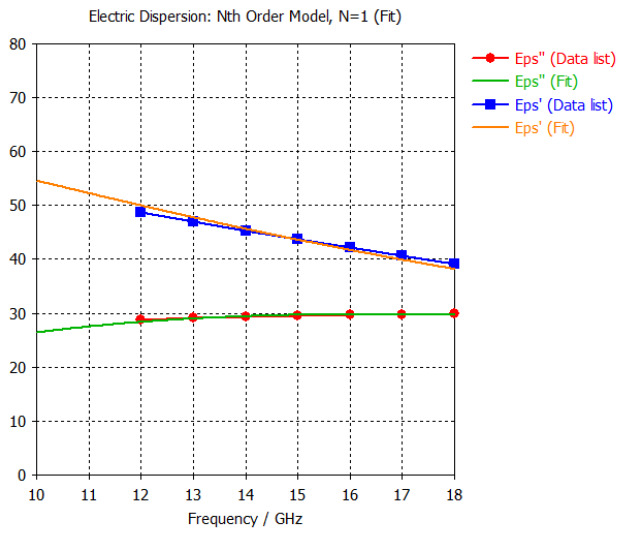
**6. Gray Matter** 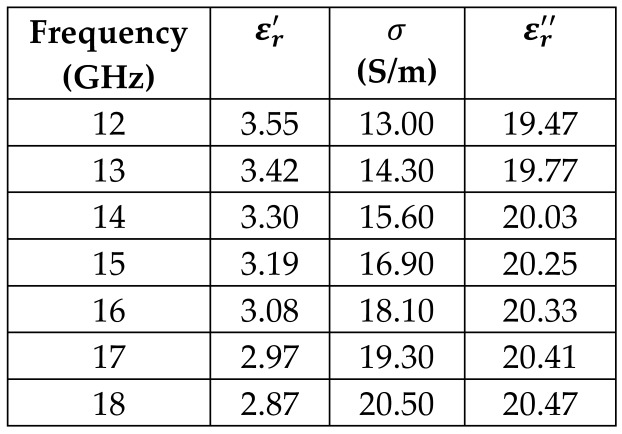	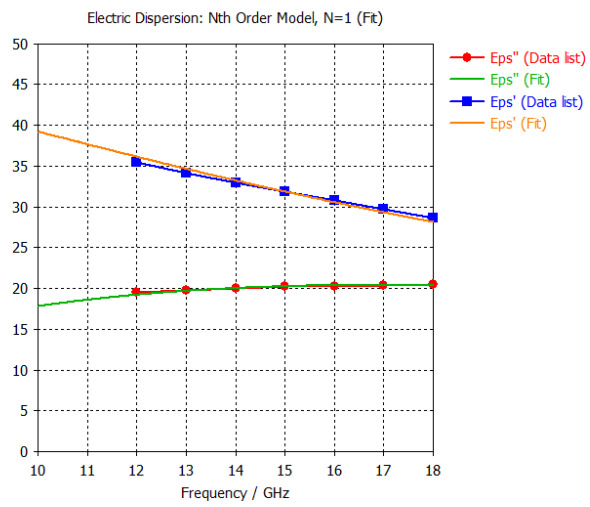
**7. White Matter** 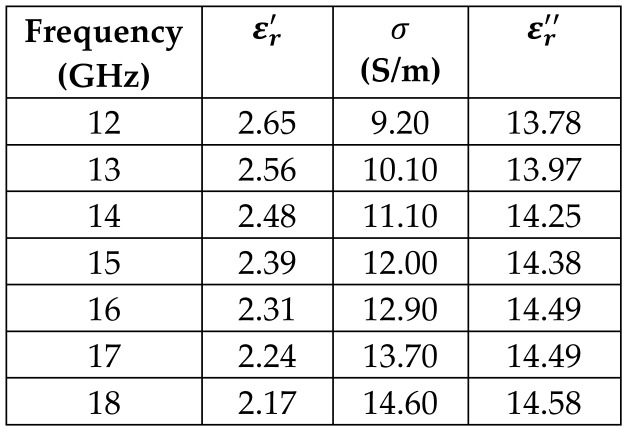	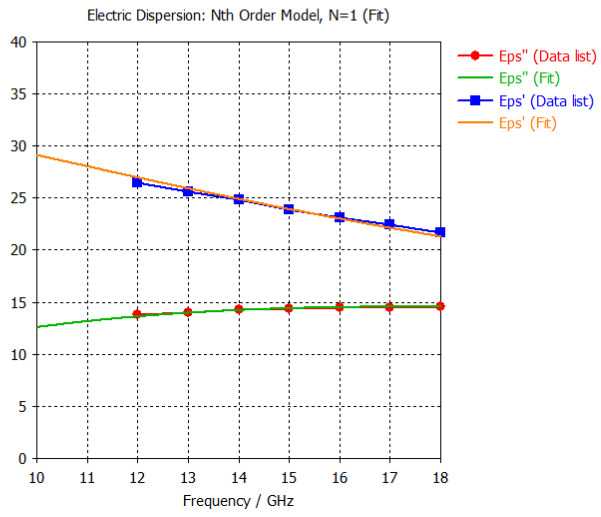

**Table 4 sensors-24-05953-t004:** Ku-band single-band antenna dimensions.

Patch Length	** *L* **	7.00 mm	Ground Length	** *L_G_* **	12.89 mm
Patch Width	** *W* **	10.13 mm	Ground Width	** *W_G_* **	12.89 mm
Inset Length	** *L_I_* **	1.00 mm	Cu Thickness (1 oz)	**H_M_**	0.035 mm
Inset Width	** *W_I_* **	0.10 mm	Substrate Thickness (Height)	**H_s_**	1.50 mm
Feed Length	** *L_F_* **	3.95 mm			
Feed Width	** *W_F_* **	0.25 mm			

**Table 5 sensors-24-05953-t005:** BW % and gains of the antenna using different substrates.

Substrate	BW%	Gain at 14.8 GHz (dBi)	Gain at 15.0 GHz (dBi)
**FR-4, ** $\varepsilon_{r}=4.3$	5.54	1.782	2.039
**Rogers 410, ** $\varepsilon_{r}=4.1$	3.87	3.958	4.331
**Rogers 430, ** $\varepsilon_{r}=4.3$	3.71	4.485	4.764

**Table 6 sensors-24-05953-t006:** S_11_ and S_21_ resonances of the proposed nine SRR metamaterials.

Resonance	S_11_	S_21_
1	6.03 GHz/−23.95 dB	5.82 GHz/−22.76 dB
2	7.83 GHz/−28.21 dB	7.40 GHz/−26.98 dB
3	9.75 GHz/−30.00 dB	9.05 GHz/−29.25 dB
4	12.05 GHz/−32.82 dB	11.07 GHz/−30.22 dB
5	14.55 GHz/−26.44 dB	13.90 GHz/−27.30 dB
6	17.43 GHz/−19.83 dB	16.77 GHz/−33.47 dB
7	20.75 GHz/−28.71 dB	18.38 GHz/−35.64 dB
8	23.00 GHz/−35.09 dB	21.54 GHz/−24.05 dB
9	-	24.15 GHz/−12.35 dB

**Table 7 sensors-24-05953-t007:** Centered S_11_ values when testing the rectangular patch using brain tumors with different diameters.

#	Tumor Diameter (mm)	Frequency (GHz)	S_11_ (dB)
	No Tumor	15.1684	−39.4112
1	2	15.1695	−40.9031
2	4	15.1790	−37.9046
3	6	15.1703	−39.4733
4	8	15.1658	−39.4792
5	10	15.1758	−37.7204
6	12	15.1575	−41.7402
7	14	15.1719	−38.7485
8	16	15.1831	−37.2366
9	18	15.1798	−37.7878
10	20	15.1923	−36.0647

**Table 8 sensors-24-05953-t008:** Centered S_11_ values when testing the disc patch using brain tumors with different diameters.

#	Tumor Diameter (mm)	Frequency (GHz)	S_11_ (dB)
	No Tumor	14.5002	−35.6476
1	2	14.4869	−36.6056
2	4	14.5083	−35.4346
3	6	14.4645	−38.7742
4	8	14.4802	−36.4792
5	10	14.4886	−36.0421
6	12	14.4822	−36.4104
7	14	14.5070	−35.7471
8	16	14.4890	−36.5539
9	18	14.5147	−34.8586
10	20	14.5097	−35.2539

**Table 9 sensors-24-05953-t009:** S_11_ values obtained at the two bands and using the two-point mapping system when testing the dual-band patch using brain tumors with different diameters.

#	Tumor Diameter (mm)	S_11_ (dB) at 13.24 GHz 1st Passband	Order 1st Point	S_11_ (dB) at 15.80 GHz 2nd Passband	Order 1st Point	Two-Point Mapping
	No Tumor	−20.1544	** *8* **	−9.0812	** *8* **	**8-8**
1	2	−20.4591	** *11* **	−9.1712	** *11* **	**11-11**
2	4	−20.0978	** *7* **	−9.0614	** *7* **	**7-7**
3	6	−20.4403	** *10* **	−9.1698	** *10* **	**10-10**
4	8	−20.1989	** *9* **	−9.0934	** *9* **	**9-9**
5	10	−19.6468	** *3* **	−8.9361	** *3* **	**3-3**
6	12	−19.2607	** *2* **	−8.8319	** *2* **	**2-2**
7	14	−19.8125	** *5* **	−8.9905	** *5* **	**5-5**
8	16	−19.6936	** *4* **	−8.9436	** *4* **	**4-4**
9	18	−20.0601	** *6* **	−9.0487	** *6* **	**6-6**
10	20	−19.1506	** *1* **	−8.8073	** *1* **	**1-1**

**Table 10 sensors-24-05953-t010:** S_11_ values obtained for the three bands and using the three-point mapping system when testing the tri−band antenna using brain tumors with different diameters.

#	Tumor Diameter (mm)	S_11_ (dB) at 13.72 GHz 1st Passband	Order 1st Point	S_11_ (dB) at 15.1095 GHz 2nd Passband	Order 2nd Point	S_11_ (dB) at 17.48 GHz 3rd Passband	Order 3rd Point	Three-Point Mapping
	No Tumor	−32.5696	** *5* **	−21.6219	** *2* **	−6.2831	** *7* **	**5-2-7**
1	2	−35.1152	** *8* **	−21.6594	** *3* **	−6.4408	** *8* **	**8-3-8**
2	4	−31.9537	** *2* **	−21.7093	** *4* **	−5.9660	** *1* **	**2-4-1**
3	6	−32.4256	** *4* **	−21.7164	** *5* **	−6.0308	** *3* **	**4-5-3**
4	8	−35.0670	** *7* **	−21.7431	** *6* **	−6.1400	** *4* **	**7-6-4**
5	10	−32.1510	** *3* **	−21.6124	** *1* **	−6.0075	** *2* **	**3-1-2**
6	12	−31.4255	** *1* **	−21.7781	** *7* **	−6.1860	** *6* **	**1-7-6**
7	15	−34.4348	** *6* **	−21.8409	** *8* **	−6.1749	** *5* **	**6-8-5**

## Data Availability

The original contributions presented in the study are included in the article, further inquiries can be directed to the corresponding author.

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
