# Peer review of "Multiple-Point Metamaterial-Inspired Microwave Sensors for Early-Stage Brain Tumor Diagnosis"

_sensors, 2024, doi:10.3390/s24185953_

Round 1
Reviewer 1 Report
Comments and Suggestions for Authors
please refer to the report in the PDF file.

Comments on the Quality of English Languagegood
Author Response
Thank you very much for your suggestions. We revised the article to address your comments:
1. The following sentences were added on page 11 to provide the S11 parameter definition.
‘The reflection coefficients or S11, the ratio of a reflected wave to an incident wave, measuring how much a wave is reflected from the studied models, are mainly used in this work.’
2. The two suggested references were added as Refs [34-35] to provide additional information on metamaterials and split ring resonators in the 1st paragraph of Section 3.3, on page 16:
‘The design strategy was to first create a multi-band enhancer, using a well-known metamaterial split ring resonator (SRR) [27-35], ...’
[34] H. Li, S. Li, H. Liu and X. Wang, Analysis of electromagnetic scattering from
plasmonic inclusions beyond the quasi-static approximation and applications, ESAIM:
Math. Model. Numer. Anal., 53 (2019), no. 4, 1351–1371, 2019.
[35] H. Li and H. Liu, On anomalous localized resonance and plasmonic cloaking
beyond the quasistatic limit, Proceedings of the Royal Society A, 474: 20180165,
http://doi.org/10.1098/rspa.2018.0165.
Reviewer 2 Report
Comments and Suggestions for Authors
What is the focus of this paper? This paper includes too many contents and thus it should be reorganized very carefully to make it understood easily.
Author Response
What is the focus of this paper? This paper includes too many contents and thus it should be reorganized very carefully to make it understood easily.
Answer: Thank you for your comments and suggestions.
The paper studies multiple-point metamaterial-inspired microwave sensors for early-stage brain tumor diagnosis. A 7-layered head phantom was used to mimic the actual human head. Since a single band detecting point is not adequate to classify the nonlinear tumor and head model parameters, dual-band and tri-band metamaterial-inspired antennas are applied to solve the problem. Both 2-point and 3-point mapping show advantages in characterizing the relationship between the S11 value and tumor size. Thus, the multi-detecting point technique can be applied to a sensor for nonlinear property
targets.
The following sentences were added as the last paragraph of the introduction section on page 3 to provide a clear content direction of this article.
‘The content direction of this article begins with the discussion of the meningioma brain tumor and human head model (Section 2), followed by the design of four types of Ku band antenna (Section 3). Then, the model designed in Section 2 is tested with the 4 antennas, starting with single-band rectangular and disc patches, and continuing with a dual-band and a tri-band antenna in Section 4, which includes simulation results and discussion.’
Reviewer 3 Report
Comments and Suggestions for Authors
The manuscript submitted to the SI describes interesting approach to the detection of early-stage brain tumor diagnostics. The subject is very interesting and relevant to the submitted journal. It is a hot spot of the present day research efforts – to designed instrumentation with very high sensitivity and redust radiation effects for the early-stage tumor detection. With this respect it would be very important to compare existing devices, laboratory prototypes or simply proposed approaches or just models in order to confirm the validity and significance of the new proposed direction. Unfortunately, the Introductions suffers a significant number of white spots with this respect. There were many papers published and prototypes presented for microwave detection of the set of magnetic nanoparticles associated with model tumor tissue or thrombosis related coagulate using a magnetic field sensor (Llandro, J.; Palfreyman, J.J.; Ionescu, A.; Barnes, C.H.W. Magnetic biosensor technologies for medical applications: A review. Med. Biol. Eng. Comput. 2010, 48, 977–998; Uchiyama, T.; Mohri, K.; Honkura, Y.; Panina, L.V. Recent advances of pico-Tesla resolution magneto-impedance sensor based on amorphous wire CMOS IC MI Sensor. IEEE Trans. Magn. 2012, 48, 3833–3839; Melnikov, G.Y.; Lepalovskij, V.N.; Svalov, A.V.; Safronov, A.P.; Kurlyandskaya, G.V. Magnetoimpedance thin film sensor for detecting of stray fields of magnetic particles in blood vessel. Sensors 2021, 21, 3621, and many others). There is no simple mentioning of this kind of sensors in the Introduction. Work is very modestly referenced for the 24 pages contributing paper. Broader discussion including the future trends and expected directions of the development could be useful for the wide audience of Sensors MDPI.
Unfortunately, no real testing of the approach are not given in the manuscript. Even most simple comparison of the experimental data and theory would significantly increase the quality of the manuscript. At least, it should be clearly written at the end of the manuscript that there were no tests neither with animals nor with humans. In addition, proper estimation of all experimental errors (no error bars are shown) must be provided. One of the problems is non-uniform distribution of the temperatures over the simulating part of the brain. How this point was taken into account?
In general, microwave devices require careful calibration and very accurate consideration of different kinds of contributing components of noise. This problem must be discussed and explained in the text. To what extent the proposed metamterial is difficult to fabricate, what is the expected degree of the repetition rate for each particular device? What is the energy/cost efficiency of proposed device (may be in comparison with other existing or proposed examples)?
Comments on the Quality of English LanguageExtra proof-reding would be an advantage.
Author Response
Thank you very much for your suggestions and comments. Please find our responses below:
1. We are grateful for your insight on the existence of microwave detection sensors.
The following sentences were added in our revision in the 1st paragraph in the Introduction section on page 2:
‘There are several works including presented prototypes for microwave detection of the set of magnetic nanoparticles associated with model tumor tissue or thrombosis related coagulation, using a magnetic field sensor [3-5]. A review of magnetic nanotechnology approach [3] for various types of sensors used to detect magnetic labels was presented. Sensitive micromagnetic sensors, referred to as MI sensors, based on magneto-impedance effect in amorphous wires and CMOS IC electronic circuits providing a sharp-pulse excitation, were developed [4]. Magnetoimpedance thin film sensors for detecting stray fields of magnetic particles in blood vessel were also proposed [5]. These techniques could be incorporated in brain tumor diagnosis.’
The suggested references were also cited as refs. [3-5].
2. The following sentences were added in the conclusion section, on page 26, to state that there were no tests either on animals, or humans:
‘This is a preliminarily simulation work on designing alternative microwave sensors using Ku band antennas. Further tests on animals or humans will be a part of critical future work.’
3. Thank you very much for pointing out the non-uniform temperature distribution.
The following sentences were added to Section 2 on page 6 in the manuscript to stress the importance of the non-uniform temperature distribution.
‘In this study, the brain layer temperatures were set constantly and uniformly. However, it is worthy to observe the effects of the non-uniform temperature distribution of the brain parts. Different temperature values assigned case by case can be one of future detailed parametric studies.’
4. In general, microwave devices require careful calibration and very accurate consideration of different kinds of contributing components of noise. This problem must be discussed and explained in the text.
Answer: Thanks to the Reviewer for pointing this out. The following sentences were added in the end of the 1st paragraph on page 2 in the revision.
‘It is important to stress that microwave devices typically require careful and precise calibration based on their sensitivity, as well as potential high noise dissipation from different components.’
5. To what extent the proposed metamaterial is difficult to fabricate, what is the expected degree of the repetition rate for each particular device?
Answer: The metamaterial structure was designed to be simple. The resolution or the smallest dimensions, i.e., linewidth and spacing, are set at 0.2 mm, which is feasible in nowadays PCB fabrication technology. A typical low-cost PCBs, FR-4 substrate, were used in both metamaterial and antenna designs.
6. What is the energy/cost efficiency of proposed device (may be in comparison with other existing or proposed examples)?
Answer: This proposed microwave detection device will be low-cost, below a thousand USD. The PCB works of the antenna and metamaterial structures are less than a hundred USD. The main portion of the total cost is for the portable vector network analyzer. However, this is a preliminarily design. When other microcontrollers and electronic parts are added, the total cost can reach a couple of thousand USD. Based on its simple parts and almost instantaneous detection, this proposed sensor has a very low energy consumption.
The service cost for other brain imaging, e.g., positron emission tomography (PET), magnetic resonance imaging (MRI), magnetic resonance spectroscopy (MRS), computed tomography (CT), and single-photon emission computed tomography (SPECT), are typically in between $1,600 - $8,400, while the MRI machine price is in between $225,000-$500,000.
Thanks to the Reviewer for commenting on this cost efficiency perspective. The proposed device is also a good alternative cost-wise for an early-stage detection.
The following sentences were added the 2nd paragraph on page 2 in the revision to emphasize the cost efficiency perspective.
‘The brain imaging machines are very expensive (e.g. an MRI machine costs in between $225,000-$500,000), leading to high service costs of approximately $1,600-$8,400. Hence, an alternative low-cost brain tumor detection device will be helpful for an initial state diagnosis.’
Round 2
Reviewer 2 Report
Comments and Suggestions for Authors
No more comments
Reviewer 3 Report
Comments and Suggestions for Authors
Submitted manuscript is interesting and in a revised version Authors made all requested changes. It can be accepted in the present state and I recommend it for special promotion.